# Impact of the COVID-19 pandemic on remote mental healthcare and prescribing in psychiatry: an electronic health record study

Rashmi Patel [1,2] Jessica Irving [1] Aimee Brinn,[1] Matthew Broadbent,[3] Hitesh Shetty,[3] Megan Pritchard,[4] Johnny Downs [5] Robert Stewart,[4] Robert Harland,[2] Philip McGuire[1,2]

► Prepublication history and additional materials for this paper is available online. To view these files, please visit the journal online (http://dx.doi.org/10.1136/bmjopen-2020-046365).

RP and JI contributed equally.

**Correspondence to**
Dr Rashmi Patel;
rashmi.patel@kcl.ac.uk

## ABSTRACT

**Objectives** The recent COVID-19 pandemic has disrupted mental healthcare delivery, with many services shifting from in-person to remote patient contact. We investigated the impact of the pandemic on the use of remote consultation and on the prescribing of psychiatric medications.

**Design and setting** The Clinical Record Interactive Search tool was used to examine deidentified electronic health records of people receiving mental healthcare from the South London and Maudsley (SLaM) NHS Foundation Trust. Data from the period before and after the onset of the pandemic were analysed using linear regression, and visualised using locally estimated scatterplot smoothing.

**Participants** All patients receiving care from SLaM between 7 January 2019 and 20 September 2020 (around 37 500 patients per week).

**Outcome measures** (i) The number of clinical contacts (in-person, remote or non-attended) with mental healthcare professionals per week.
(ii) Prescribing of antipsychotic and mood stabiliser medications per week.

**Results** Following the onset of the pandemic, the frequency of in-person contacts was significantly reduced compared with that in the previous year (β coefficient: −5829.6 contacts, 95% CI −6919.5 to −4739.6, p<0.001), while the frequency of remote contacts significantly increased (β coefficient: 3338.5 contacts, 95% CI 3074.4 to 3602.7, p<0.001). Rates of remote consultation were lower in older adults than in working age adults, children and adolescents. Despite this change in the type of patient contact, antipsychotic and mood stabiliser prescribing remained at similar levels.

**Conclusions** The COVID-19 pandemic has been associated with a marked increase in remote consultation, particularly among younger patients. However, there was no evidence that this has led to changes in psychiatric prescribing. Nevertheless, further work is needed to ensure that older patients are able to access mental healthcare remotely.

## INTRODUCTION

The COVID-19 pandemic caused by SARS-CoV-2, and accompanying physical distancing and travel restrictions, has had a tremendous impact on the delivery of healthcare worldwide. It has particularly affected mental healthcare, making it more difficult for people to be seen in-person and harder to deliver treatments.[1–3] The pandemic has also increased the exposure of patients to factors known to exacerbate mental health disorders, including isolation, unemployment, inactivity and decreased social support.[4] The extent to which these changes have impacted upon mental health is unknown,[5] but is critical to the ongoing reconfiguration of services in response to the current pandemic.[4]

The recent advent of technology to support remote communication in healthcare (telemedicine)[6] has enabled a shift from in-person to remote clinical contact in response to the COVID-19 pandemic. This has facilitated healthcare following the adoption of travel and physical distancing restrictions originally

### Strengths and limitations of this study

► The use of electronic health record data allowed rapid analysis of the associations of the COVID-19 pandemic with remote mental healthcare in a large sample of patients.
► Natural language processing tools enabled access to free text data on medication prescribing in electronic health records.
► The large sample size increases the robustness of our findings and generalisability to real-world clinical practice.
► A short follow-up period precluded the investigation of associations between remote consultation, diagnoses and clinical outcomes.
► The study analysed secondary mental healthcare data which may not have included information on psychiatric prescribing in primary care.

developed by the UK government in response to the spread of SARS-CoV-2.[7]

Prior to the COVID-19 pandemic, the application of telemedicine in mental health has been limited, and little is known about the clinical and sociodemographic factors that may influence the effectiveness of remote consultation in this context.[8] As the current pandemic evolves, while travel and physical distancing restrictions are expected to continue,[9–11] remote consultation is likely to constitute a major component of mental healthcare until the pandemic is over. Furthermore, it is possible that in the absence of a pandemic, remote consultation could help to improve access to and reduce costs of delivering community mental healthcare.[7] Thus, it is critical to determine the impact, potential benefits and disadvantages of remote consultation in mental healthcare.

Electronic health record (EHR) studies provide an opportunity to look at patterns of healthcare service delivery across populations.[12] Natural language processing (NLP) techniques can be used to enrich EHR datasets by extracting clinical relevant data from assessments, reviews and correspondence documented as unstructured free text.[13–16] Web-based data visualisation platforms can then be used to present population data derived from EHRs to support healthcare policy making.[17 18]

In the present study, we analysed EHR data from a large provider of mental healthcare in South London (UK) to assess the impact of the current COVID-19 pandemic on rates of remote consultation and prescribing of psychiatric medications.

## METHODS
### Participants and setting
We extracted data from de-identified EHRs for individuals accessing secondary mental healthcare within the South London and Maudsley (SLaM) NHS Foundation Trust from 7 January 2019 to 20 September 2020. SLaM holds EHRs for over 450 000 people who have received mental healthcare since 2007.[12] The Trust provides community and inpatient mental healthcare within its catchment area which includes the London boroughs of Lambeth, Southwark, Croydon and Lewisham. Its service provision covers the following specialty groupings: Addictions; Behavioural and Developmental Psychiatry; Child and Adolescent Mental Health Services; Mental Health of Older Adults and Dementia; Mood, Anxiety and Personality; Psychological Medicine, and Psychosis. Its services are structured into three age groups: children and adolescents (under 18 years), working age adults (18–64 years) and older adults (65 years plus).[12] SLaM's catchment area varies considerably in terms of ethnic composition, education, urbanicity and area-level deprivation.[12 19] Overall the SLaM catchment boroughs are representative of London as a whole in terms of age, gender, education and socioeconomic status.[19]

### Data extraction
We extracted separate datasets for each week from January 2019 to September 2020. We chose this time period to provide a whole calendar year of data representing typical clinical activity prior to the onset of the COVID-19 pandemic, which was declared by WHO in March 2020.[20] The cohort for each week included all patients with an active referral to SLaM, defined as having been accepted to a SLaM community or inpatient service on, or before the Sunday of each week and having been discharged on, or after the Monday of the same week (for those patients discharged from SLaM clinical services prior to September 2020). This method was chosen to enable assessment of changes in the frequency of data in the context of the number of active patients within a given week, taking into account fluctuations in rates of acceptance to, and discharge from clinical services during the period of data analysis, and the fact that some patients may have had an active referral to SLaM for a duration of less than a week (eg, patients who receive a single assessment but are then discharged without further follow-up).

The Clinical Record Interactive Search (CRIS) tool was used to extract data. CRIS is a bespoke clinical informatics platform and governance framework to support an electronic mental health case register whose infrastructure is based on Microsoft Structured Query Language (SQL) and provides an automated pipeline for EHR data deidentification, database assembly and query to facilitate analyses of EHR-derived data of patients receiving care from SLaM.[12 19] The source EHR data are generated by SLaM clinicians, including psychiatrists, psychologists, nurses and other health and social care professionals who document clinical assessments, reviews and progress in the provision of mental healthcare in the community or psychiatric hospitals. Access to data is restricted to honorary or substantive employees of SLaM.

### Patient and public involvement
The project was approved by the local SLaM CRIS Oversight Committee (chaired by a service user representative) who review and approve applications to extract data for research. The CRIS infrastructure was designed and is managed with ongoing service user input to ensure all research projects comply with data governance, legal and ethical guidelines. Further details are available on the SLaM CRIS website.[21]

### Defined variables
Data were extracted on the number of active patients at any point in a given week (referred to as 'patients registered with SLaM services'), age, gender, the number of contacts with mental healthcare professionals that were attended in-person, remotely or where a patient did not attend (DNA), and psychotropic medications (antipsychotic drugs and mood stabilisers).

Data on contacts with mental healthcare professionals were obtained from the *Events* input field in the EHR. Clinicians use the Events field to record the content and

outcome of clinical appointments with patients. When an appointment for the patient to be assessed or reviewed by a clinician is scheduled, the clinician (or a healthcare service administrator) will create an *Event* and record the appointment date and time, whether the appointment was attended by the patient, and the Event Type (ie, in-person or a remote contact). Data on contacts with mental healthcare professionals were defined as follows:

1. In-person contacts: appointments attended by a patient and clinician recorded as a 'Face To Face' or 'Group Contact' Event Type. The 'Face To Face' Event Type refers to an appointment which is conducted with both the patient and clinician present in the same physical space which could be at a mental health service clinic/outpatient department or an alternative location such as the patient's own home. The 'Group Contact' Event Type refers to an appointment conducted in a group setting within the same physical space (ie, multiple patients and one or more clinicians, eg, as part of group psychological therapy).

2. Remote contacts: defined as appointments attended by a patient and clinician recorded as a 'Phone' or 'Video (virtual) appointment' Event Type. We did not analyse data on written forms of remote contact between patients and clinicians (recorded as 'Email', 'Letter', 'Mail' or 'Short Message Text' Event Type) as these methods of contact are not necessarily attended by the patient and clinician contemporaneously and the date of the recorded Event may not correspond with the date of the written communication being received by the patient.

3. DNA contacts: defined as any unplanned appointment cancellation (in-person or remote).

Age was determined according to the age of each active patient on the Sunday of each week and was categorised in groups defined as under 18 years, 18–64 years and 65 years and over. These groups were chosen to reflect the age group structure of SLaM clinical services which provide care for children and adolescents, working age adults and older adults, respectively.[12]

We extracted recorded mentions of antipsychotics and mood stabilisers using two sources: structured input fields within the EHR to record prescribed psychotropic medications and NLP tools applied to free text EHR data including documentation of clinical assessments, and progress notes, clinical reports and correspondence between clinicians. The development of these NLP tools and example use cases have been previously described[12 16 22 23]; details on manual validation can be found on the CRIS website.[24]

Antipsychotics were defined according to the Maudsley Guidelines for Advanced Prescribing in Psychosis[25] and grouped as oral antipsychotics and long-acting injectable (LAI) depot antipsychotics. A further analysis of the most frequently prescribed antipsychotics was conducted including amisulpride, aripiprazole, clozapine, haloperidol, lurasidone, quetiapine, olanzapine, risperidone, sulpiride, zuclopenthixol, aripiprazole LAI, paliperidone

LAI (including both 1-monthly and 3-monthly preparations), haloperidol LAI, risperidone LAI and zuclopenthixol LAI. Non-antipsychotic mood stabilisers were defined as carbamazepine, lamotrigine, lithium and valproate. The analyses focused on antipsychotics and mood stabilisers because these are the most frequently prescribed psychotropic medications in UK secondary mental healthcare. We chose not to analyse data on antidepressants, as these are often initiated in primary care and these data are not comprehensively represented in the secondary mental healthcare dataset analysed in this study.

### Data analysis and visualisation

Data were cleaned and visualised using Exploratory V.6.2.2 (https://docs.exploratory.io/), a user interface frontend to R 4.0.x for data analysis and visualisation. Results were communicated via an interactive dashboard which enables users to zoom and scroll within charts to highlight specific data points or series. This platform also facilitates updates to the dataset over time beyond the date of publication of this article and enables rapid visualisation of data to inform future healthcare policy. A brief guide on how to visualise data from the present study (available at http://rpatel.co.uk/TelepsychiatryDashboard) is provided in the online supplemental material.

Based on the declaration of the COVID-19 pandemic in March 2020, we chose an index period of Monday 4 March 2019 to Sunday 1 March 2020 against which to compare subsequent data. We estimated the mean and 95% CI for all weeks within the index period for each analysed variable. We performed separate univariate linear regression analyses to compare the weekly counts for each variable in each of the calendar months of March, April, May, June, July, August and September 2020 against the mean weekly counts in the index period. The β coefficient of each analyses represents the estimated difference from the index period mean for each variable within each analysed month. This method of monthly comparisons was chosen to enable an estimation of changes in frequency of each variable after the onset of the pandemic while taking into account weekly variations which would have occurred due to the non-working days of Easter in April and the UK bank holidays during May.

Data were aggregated and presented in charts using Exploratory software. Each data point in a given chart represents the count for the analysed variable on the Sunday of each week. Locally estimated scatterplot smoothing (LOESS) was used to superimpose a smoothed trendline (represented by a dotted line and shaded area for the CI). LOESS was used to smooth out variation between each week and highlight data points for weeks indicating significant variation away from the lower or upper bound of the CI. In order to effectively visualise data in limited screen space, the y-axes of charts were adjusted to fit the majority of the data points within the viewable area for each chart. When viewing the results, care should be taken to examine the y-axis to determine

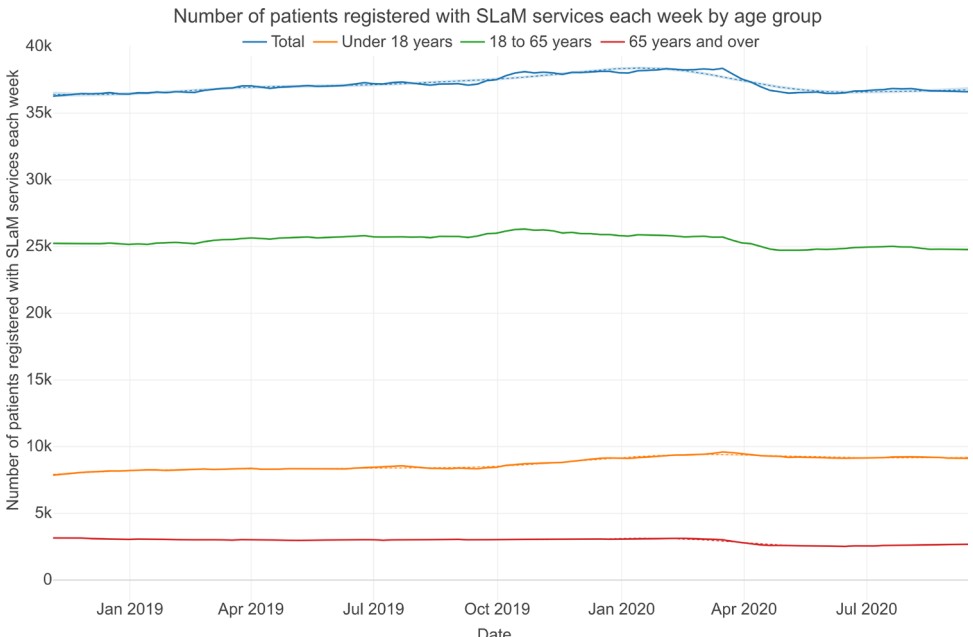

Number of patients registered with SLaM services each week by age group
— Total — Under 18 years — 18 to 65 years — 65 years and over

**Figure 1** Number of patients registered with SLaM services each week by age group. SLaM, South London and Maudsley.

scale and intercept prior to interpreting data points and fitted LOESS curves within the charts. Patients with missing age and gender data were excluded from stratified age and gender charts respectively but were included in all non-stratified charts.

## RESULTS

### Patients registered with mental health services

A mean number of 37 563 patients (95% CI 37 420 to 37 707) were registered with clinical services each week during the index period. Regression analyses indicated consistent decreases for each month of April 2020 to September 2020 in the number of patients registered per week relative to the index period, with the exception of registration for those under 18 years, which showed the opposite trend (figure 1 and online supplemental table 1 and figure 1). However, the relative changes in the total number of patients registered with SLaM services during each week after April 2020 were modest, representing a maximum change of around 2% following the index period (figure 1 and online supplemental figure 1). The number of patients newly registered with SLaM services each week dropped between March and April followed by a modest recovery in May (online supplemental figure 2).

### Contacts with mental health professionals

The number of weekly contacts reduced considerably between December 2019 and January 2020, reflecting a slowdown in clinical activity during the Christmas and New Year period (figure 2). From March 2020, in-person contacts reduced substantially from around 9000 per week to 3000 per week in early April 2020, corresponding with the implementation of travel and physical distancing restrictions. Over the same period there was a substantial increase in remote contacts from around 2500 per week

in early March 2020 to around 8000 per week by the end of April 2020 (figure 3), while the total number of clinical contacts per week dropped from around 12 500 in mid-March to around 10 000 in mid-April 2020 (figure 2). Before March 2020 very few video consultations were conducted. Following March 2020, the frequency of video consultations increased to around almost 30% of all remote consultations by July 2020 (online supplemental figures 3 and 4).

These visual trends were confirmed through linear regression analyses, which found consistent decreases in mean weekly in-person contacts each month from March to September relative to the previous year. This was matched by consistent increases in remote contacts compared with the previous year. The number of unattended appointments temporarily reduced in April, May, June and September 2020 (online supplemental table 1).

It is notable that while there were reductions in clinical contacts during the early phase of the COVID-19 pandemic, these were smaller than those observed between December 2019 and January 2020, prior to the pandemic.

Visual inspection of figure 4 illustrates the breakdown of clinical contacts per week by age. Children and adolescents under 18, working age adults and older adults show varying degrees of reduction in weekly contacts following the onset of the COVID-19 pandemic. The total number of clinical contacts for children and adolescents between mid-March and mid-May reduced to a lesser degree than for working age adults and older adults.

Visual inspection of figure 5 indicates a compensatory increase in remote consultations for children and adolescents, with a less pronounced increase seen in patients of working age and in older adults. Although there was no clear difference between number of weekly clinical

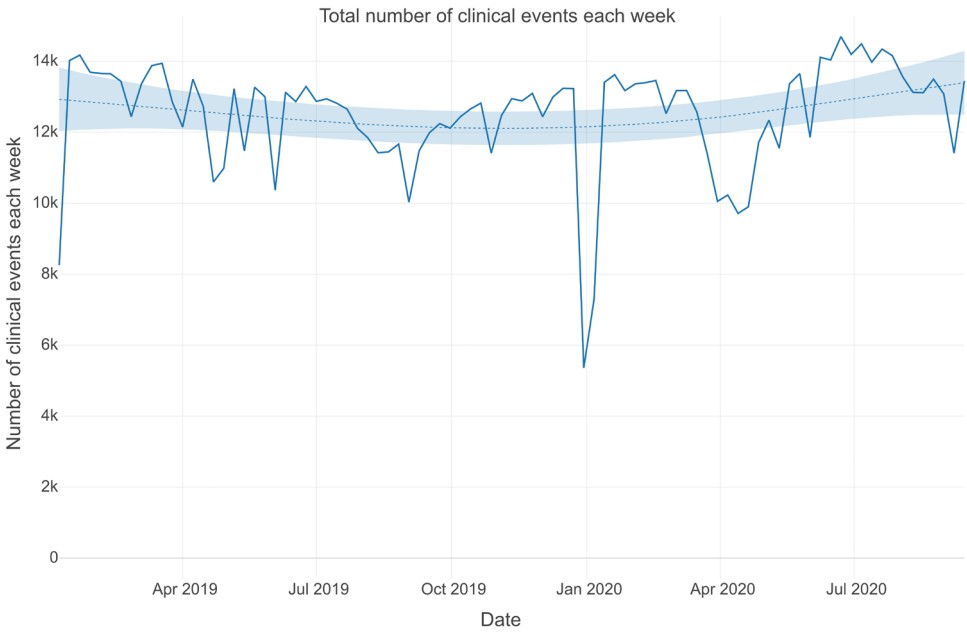

Figure 2 Total number of clinical events each week.

events by gender (online supplemental figure 5), there was a greater increase in remote consultations following March 2020 in female than male patients (online supplemental figure 6).

## Antipsychotics

Visual inspection of antipsychotic reporting trends yielded two main findings: first, there was no evidence of a substantial change in total antipsychotic mentions in EHRs following the onset of the pandemic (for either oral or LAI depot antipsychotics). Second, there was a large decrease in mentions of antipsychotics between December 2019 and January 2020 (online supplemental figure 7). Free text mentions of antipsychotics outnumbered mentions in structured input fields by a factor of roughly three (online supplemental figure 8). When grouped by age, antipsychotic mentions reduced during 2019, before increasing towards January 2020 in children and adolescents. Working age adults showed a reduction in antipsychotic mentions during 2019, with a nadir between December 2019 and January 2020. In contrast, antipsychotic mentions steadily increased among older adults over 65 (online supplemental figure 9). There were no clear differences in antipsychotic mentions by gender (online supplemental figure 10).

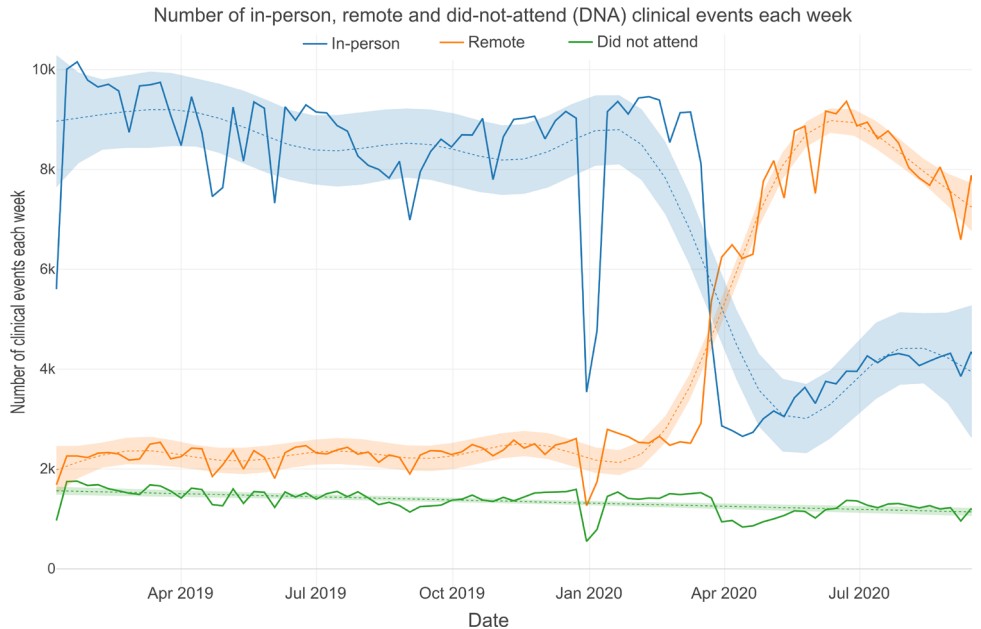

Figure 3 Number of in-person, remote and DNA clinical events each week.

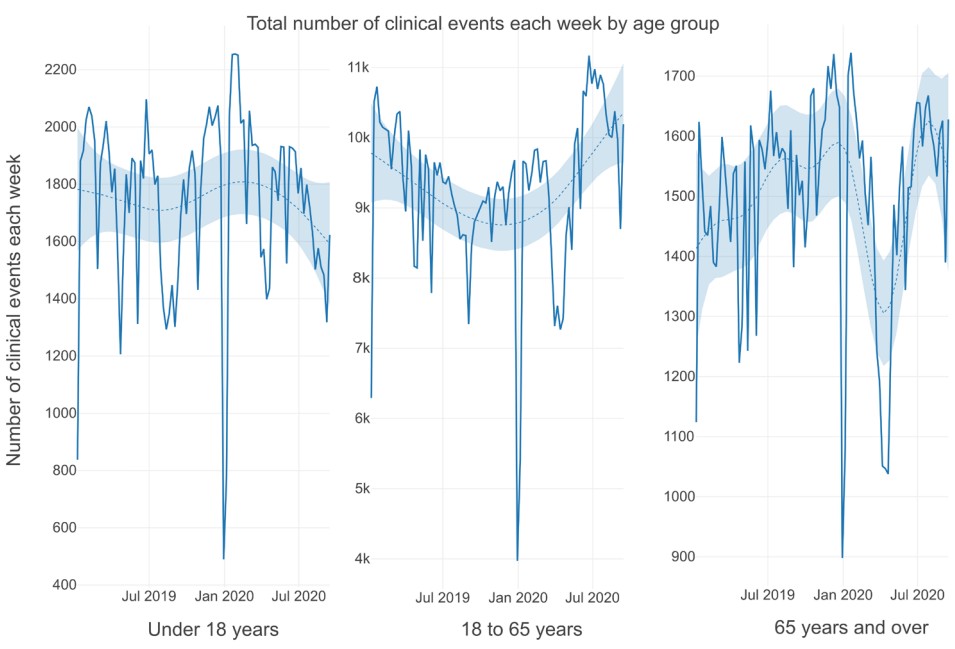

**Figure 4**  Total number of clinical events each week by age group.

There was a statistically significant and consistent increase in weekly aripiprazole depot mentions in the months of March 2020 to September 2020 compared with mean rates across the index period. Risperidone depot mentions were reduced in March, June, August and September 2020 compared with the index period. A statistically significant increase in clozapine mentions was observed in April only ($\beta=79.2$, $p<0.001$) compared with the previous year (online supplemental table 1).

### Mood stabilisers
Total mentions of mood stabilisers slowly declined from the beginning of the observation period in January 2019

(online supplemental figures 11, 12 and 13). In terms of individual mood stabilisers, from January 2019, lithium mentions increased and mentions of valproate and lamotrigine decreased (online supplemental figure 11). Regression analysis found small decreases of marginal significance in carbamazepine and lamotrigine mentions between April 2020 and September 2020 relative to the pre-pandemic index period (online supplemental table 1).

### DISCUSSION
We found that the onset of the COVID-19 pandemic was associated with a rapid and major shift in the mode

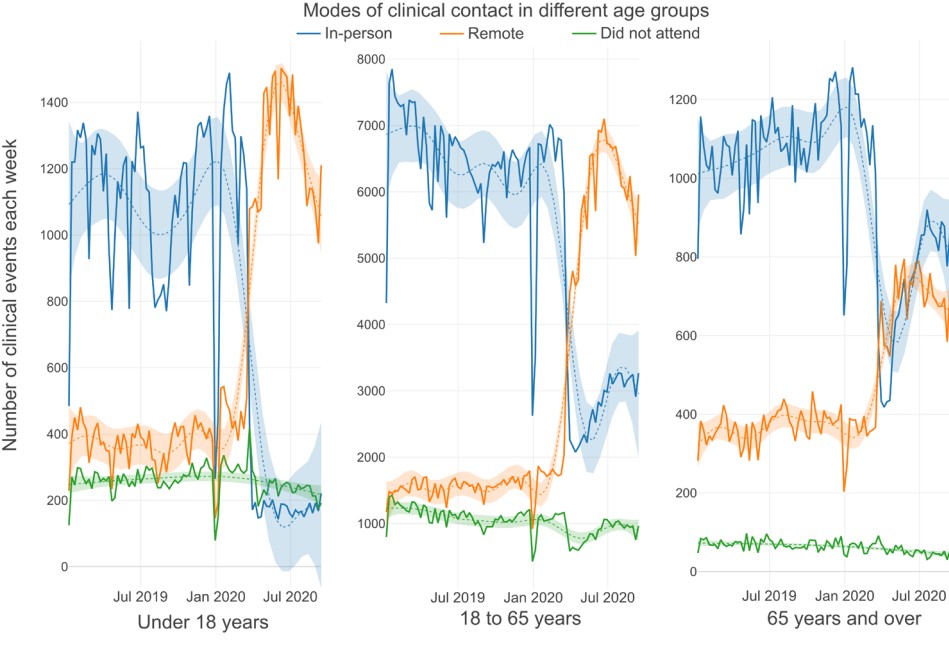

**Figure 5**  Modes of clinical contact in different age groups.

of mental healthcare consultations, from in-person to remote in the early phase (between March and May 2020), followed by a gradual shift back towards in-person consultations over the following months. Reassuringly, these substantial changes do not appear to have been associated with dramatic shifts in the clinical activity of mental health services, as measured by the number of mentions of psychotropic medications in free text EHRs. A compensatory increase in remote mental health consultation in response to the COVID-19 pandemic mirrors similar developments within physical healthcare[26] and is in keeping with a previous analysis of working age adults using the same dataset.[27]

The shift from in-person to remote consultation was most pronounced for children and adolescents (and their parents and carers), less marked in working age adults, and least dramatic in older adults. Mental health services for older adults have not utilised remote methods for consultation to the same degree as services for working age adults and children, and this has been associated with a reduction in total clinical contacts during the early phase of the pandemic between March and May 2020. Importantly, some forms of remote contact rely on the availability of electronic hardware (a smartphone or computer), access to a stable electricity supply and high-speed internet connection, and the ability to use the technology. To enable successful remote consultation, these resources are required by both patients and clinicians, and both groups need to be able to trust the technology (with respect to safety, reliability and privacy) and be motivated to use it.[28]

From April 2020, the number of patients registered per week in mental health services decreased relative to the period of March 2019 to March 2020. This is in keeping with an earlier analysis of the same dataset examining the period from 1 February to 31 March 2020.[29] This reduction in patients registered following April 2020 may reflect the discharge of patients in the initial stages of the pandemic who may have previously had limited input from mental health services. This is supported by the fact that there was no demonstrable change in clinical activity measured through mentions of antipsychotic and mood stabiliser medications during the same period. Nonetheless, the relative reduction in number of active patients in a given week during the early stages of the pandemic was modest, representing around 2% of the total mean number of active patients between March 2019 and March 2020.

Throughout the pandemic period, the number of mentions of antipsychotics (whether oral or depot) remained steady. The only exception was an increase in mentions of aripiprazole depot, but this may reflect its recent addition to the NHS formulary. We found a reduction in mentions of antipsychotics for children and adolescents. This may be due to a reduction in the number of inpatient psychiatric beds for children and adolescents in SLaM due to the closure of two inpatient units during the same period which could have led to reduced

documentation of antipsychotics in this patient group. We observed a steady increase in mentions of antipsychotics for older adults which predated the COVID-19 pandemic. The gradual increase in documentation may or may not reflect a change in prescribing (see the section Strengths and limitations), and could be related to service reconfigurations associated with changing frequency of clinical documentation and the increasing focus on monitoring of antipsychotic use in dementia.

Clozapine prescribing differs from that for other antipsychotics, in that it requires regular blood monitoring, making in-person contact essential. The increase in clozapine mentions observed in April 2020 relative to the previous year may be due to the introduction of emergency guidelines regarding frequency of full blood count monitoring to manage possible disruptions during the initial period of travel and physical distancing restrictions.[30] The increase in mentions of clozapine in April could reflect increased activity among clinicians creating treatment plans to navigate the challenges of managing full blood count monitoring for these patients.

Our analysis also revealed a decreasing trend in overall mood stabiliser mentions over time that was not related to the COVID-19 pandemic period. Notably, lithium mentions have increased and mentions of valproate and lamotrigine have decreased. The decreasing trend in valproate mentions may be explained by a reduction in prescriptions among female patients of childbearing age due to increased awareness of its teratogenic effects.[31 32] It may be that the decrease in valproate prescribing is associated with a compensatory increase in mentions of lithium, or this could be related to efforts to increase access to lithium for people suffering with treatment resistant depression or bipolar disorder.[33 34]

## Implications

The lack of digital technology resources among certain vulnerable groups creates barriers to accessing remote mental healthcare. People with low incomes (over-represented among people with serious mental disorders) may not be able to afford the necessary hardware/data connection.[35] Serious mental illnesses are associated with major impairments in motivation which impair the ability of affected individuals to use remote consultation technology.[36] And many people may not have had the opportunity to access training to gain skills in using digital healthcare tools.[28] Furthermore, people with serious mental illnesses are over-represented in deprived areas, which are the very populations at increased risk for uncontained COVID-19 transmission resulting in greater travel and physical distancing restrictions which would necessitate remote consultation to gain access to mental healthcare.[7 37]

Our research indicated that the incidence of missed appointments decreased as appointments changed from in-person to remote between April and June. Adherence to a treatment plan is an essential component for successful mental healthcare management; for people

diagnosed with schizophrenia, failure to attend outpatient care following admission increases risk of relapse and readmission.[38] Relapse of serious mental illness poses large additional costs the NHS: an estimate of four times greater costs for patients who relapse, compared with those who do not.[39] Considering non-attendance at mental healthcare appointments is reported to be greater than most other healthcare settings,[40 41] remote methods could be harnessed more often to reduce the financial burden of non-attended appointments.[41]

Future work must urgently identify how to remove the barriers faced in access to and ability to use digital technology for remote consultation. A Canadian study found that patients receiving telephone appointments felt able to present the same information as in-person and were equally as satisfied as patients receiving in-person appointments. However, the telepsychiatry group reported lower levels of support and encouragement than in-person patients.[42] Furthermore, a systematic review, which analysed the acceptability of online and mobile phone app interventions for serious mental illness, indicated that while hypothetical acceptability of online and mobile phone interventions was low, actual acceptability was generally high.[43] It is possible that certain groups may feel less able to communicate their mental health difficulties remotely, while others may find the opposite. At the same time, clinicians may feel less able to adequately assess the mental state of patients through remote technology compared with in-person assessment. Further in-depth, qualitative research is required to elucidate the views and experiences of different patient and clinician groups.

### Strengths and limitations

The availability of EHRs in SLaM has enabled the rapid development of a data visualisation platform to assess key metrics of mental health service delivery during a period of tremendous challenge to service provision during the COVID-19 pandemic. Our findings will support mental healthcare strategy to reduce barriers to those who face difficulties accessing remote care. This data visualisation approach could be applied to other forms of EHR derived healthcare data on a real-time basis to enable healthcare service providers and policy-makers to adapt service provision in time of crisis, and to measure the impact of changes to service provision on performance metrics such as access to remote care.

We were able to examine changes in the frequency of different types of mental health service consultation and mentions of psychotropic prescribing over time. However, it was not possible to analyse the nature of in-person and remote consultations. Remote consultations can take many forms and the ability to perform certain clinical tasks may vary depending on the modality employed. For example, email and text messaging may enable communication between patients and clinicians, but opportunities for real-time clinical assessment and review are limited. In contrast, a phone call can provide some information on history and mental state, but the availability of a video stream enables a more detailed assessment of mental state and environment. Certain clinical tasks may have been better suited to different forms of remote consultation. NLP could be used to interrogate the content of free-text documentation of remote consultations and to compare these to documentation of in-person consultations to identify barriers to performing different clinical activities and how these could be overcome.

While we are able to examine associations with age and gender, it was not possible to meaningfully investigate variation in rates and nature of remote consultation based on psychiatric diagnosis owing to the short window of data available within the present study, and because diagnostic data are not comprehensively documented at every clinical encounter. A cohort study comparing people with different psychiatric diagnoses on the frequency of in-person, remote and non-attended consultations would help to determine which groups of patients and treating clinicians have more avidly taken up remote forms of consultation during the early phase of the pandemic and what can be done to ensure digital access across all clinical subgroups.

It is important to note that changes in frequency of clinical documentation of psychotropic medications are not necessarily correlated with changes in prescribing per se, and could reflect differences in the nature and frequency of EHR documentation in different clinical settings. Clinicians document medication prescribing in structured fields and free text as part of the intended treatment plan agreed with the patient, but this does not necessarily equal the number of prescriptions being dispensed by a pharmacy or being subsequently taken by the patient. It is likely that a proportion of prescriptions are not dispensed and a proportion of those which are dispensed are not taken, and that this may vary depending on the clinical setting or patient group receiving treatment. However, it is unlikely that psychiatric medications would be prescribed or dispensed in the absence of documentation as part of a treatment plan and so if there had been increases in the rates of medication prescribing, this would be correlated with an increase in the frequency of recording in the EHR.

SLaM is a secondary mental healthcare setting. Patients with chronic serious mental disorder may, however, be predominantly managed by their general practitioner meaning their psychotropic medication prescribing may not be represented within secondary mental health EHR data. This is a consistent shortcoming in secondary care EHR research. Better linkages between primary and secondary care EHR systems at the individual patient level would help to facilitate more comprehensive analysis of psychotropic prescribing in people receiving secondary mental healthcare. The approach taken in the present study to examine psychotropic prescribing trends is only possible in healthcare settings with electronic case registers that capture prescribing data. Only ~50% of upper-middle-income and high-income countries (n=23) have adopted national EHR systems (2016 data, WHO).[44] Adoption rates of EHRs are much lower in the lower-middle-income contries (35%;

n=10) and low-income countries (15%; n=3). Adoption rates for psychiatric case registers for secondary analysis in research studies, such as that held by SLaM, are far lower.

While the present dashboard focuses on medication it will also be important to assess impacts on psychological and occupational therapies. One question is whether the shift to remote contacts is associated with changes to the delivery of psychological therapies and if certain groups of people are benefiting, or are marginalised, from remote therapies. This approach could be used to identify hard-to-reach groups and develop interventions tailored towards these groups and challenges they may face accessing remote care.

## Conclusions

In summary, our findings indicate that the COVID-19 pandemic has led to a rapid shift from in-person to remote methods of mental healthcare consultation, without a significant change in the overall level of clinical activity. In addition, we found marked differences in the application of remote consultation by patient age, with lower rates of among older adults, and the highest among children and adolescents. Given that travel and physical distancing restrictions related to the COVID-19 pandemic are likely to persist to some degree in the near future, it is important to ensure all people who receive and provide mental healthcare have access to the digital technology, training and clinical and social support required to access remote consultation, which is likely to continue to be an important modality to support mental health service delivery in the years following the pandemic.

**Author affiliations**
[1]Department of Psychosis Studies, Institute of Psychiatry, Psychology & Neuroscience, King's College London, London, UK
[2]Psychosis Clinical Academic Group, South London and Maudsley NHS Foundation Trust, London, UK
[3]Biomedical Research Centre Nucleus, South London and Maudsley NHS Foundation Trust, London, UK
[4]Department of Psychological Medicine, Institute of Psychiatry, Psychology & Neuroscience, King's College London, London, UK
[5]Department of Child & Adolescent Psychiatry, Institute of Psychiatry, Psychology & Neuroscience, King's College London, London, UK

**Contributors** The study was conceived by RP. Data extraction and statistical analysis were performed by JI supervised by RP and supported by MB and HS. Reporting of findings were carried out by RP, JI and AB. All authors (RP, JI, AB, MB, HS, MP, JD, RS, RH and PM) contributed to study design, manuscript preparation and approved the final version.

**Funding** MB, HS, MP, RS and PM have received funding from the National Institute for Health Research (NIHR) Biomedical Research Centre at South London and Maudsley NHS Foundation Trust and King's College London, which also supports the development and maintenance of the BRC Case Register. RP has received funding from a Medical Research Council (MRC) Health Data Research UK Fellowship (MR/S003118/1) and a Starter Grant for Clinical Lecturers (SGL015/1020) supported by the Academy of Medical Sciences, The Wellcome Trust, MRC, British Heart Foundation, Arthritis Research UK, the Royal College of Physicians and Diabetes UK. RS has received funding from a Medical Research Council (MRC) Mental Health Data Pathfinder Award to King's College London, an NIHR Senior Investigator Award and the National Institute for Health Research (NIHR) Applied Research Collaboration South London (NIHR ARC South London) at King's College Hospital NHS Foundation Trust.

**Competing interests** All authors have completed the ICMJE uniform disclosure form at www.icmje.org/coi_disclosure.pdf and declare: RS has received funding from Janssen, GSK and Takeda outside the submitted work. RP has received funding from Janssen, Induction Healthcare and Holmusk outside the submitted work. The other authors declare no competing interests.

**Patient consent for publication** Not required.

**Ethics approval** The dataset has received ethical approval from Oxfordshire REC C (Ref: 18/SC/0372) for secondary analysis of deidentified EHR data to support mental health research.

**Provenance and peer review** Not commissioned; externally peer reviewed.

**Data availability statement** Data are available on reasonable request. The data accessed by CRIS remain within an NHS firewall and governance is provided by a patient-led oversight committee. Access to data is restricted to honorary or substantive employees of the South London and Maudsley NHS Foundation Trust and governed by a local oversight committee who review and approve applications to extract and analyse data for research. Subject to these conditions, data access is encouraged and those interested should contact RS (robert.stewart@kcl.ac.uk), CRIS academic lead.

**ORCID iDs**
Rashmi Patel http://orcid.org/0000-0002-9259-8788
Jessica Irving http://orcid.org/0000-0002-2847-6508
Johnny Downs http://orcid.org/0000-0002-8061-295X

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
