## [Reviewer comments · BMJ Open]

ARTICLE DETAILS

TITLE (PROVISIONAL)	Impact of the COVID-19 pandemic on remote mental healthcare and prescribing in psychiatry: an electronic health record study
AUTHORS	Patel, Rashmi; Irving, Jessica; Brinn, Aimee; Broadbent, Matthew; Shetty, Hitesh; Pritchard, Megan; Downs, Johnny; Stewart, Robert; Harland, Robert; McGuire, Philip

VERSION 1 – REVIEW

REVIEWER	Anjana Rao Kavoor Monash Medical Center, Monash Health Australia
REVIEW RETURNED	20-Dec-2020

GENERAL COMMENTS	The given study is on a topic of relevance given the current global situation. It aims to investigate the impact of the COVID-19 pandemic on the use of remote consultation and on the prescribing of psychiatric medications in various age cohorts. The paper is perspicuously written and easy to follow. The study has employed a sizable data set with a sound methodology and clear presentation of results. Results have confirmed the given research question of the impact on remote consultations, as one may presume given the limited alternatives during such a pandemic. It is interesting that the study ascertained that there was no significant difference in antipsychotic prescription during these consultations, although, like the authors rightly pointed out, this is only a part of the clinical interaction. It would also have been interesting to discern how many of those remote consultations were done over email or letters as opposed to video calls. Given the limitations that the authors have already delineated, it is a well conducted study.
--

REVIEWER	Milou Feijt Eindhoven University of Technology
REVIEW RETURNED	27-Dec-2020

GENERAL COMMENTS	I had the pleasure to review your article. I think it is an interesting paper, especially because of the access to the great amount of data and the novel way of presenting this visually. I do have some comments, which I will elaborate upon below: - As you can see from the checklist point 1 (and related 3 and 9), my main concern is about the definition of a clear and specific research question and study objective. To my opinion, it is well-defined in the abstract: "We investigated the impact of the pandemic on the use of remote consultation and on the prescribing of psychiatric medications.", as this is precisely what you have done. However, the
--

study is introduced differently in the introduction, with a much less explicit research question/objective. "we analysed EHR data from a large provider of mental healthcare in South London (UK) to assess the impact of adopting remote consultation in response to the current COVID-19 pandemic. We tested the hypothesis that moving to remote consultation could be achieved without compromising the delivery of key components of secondary mental health care."

As a reader, this raises questions, as the objective is now formulated in much more general terms; it makes me wonder: what do you refer to with "the impact" and "key components of secondary mental health care". This is confusing, but more important, it also promises something that you can/do not deliver in the rest of the paper. I would advise to match the formulation in your introduction to the one in your abstract. This will also largely solve points 3 and 9 on the checklist, as your study design and results do match this more specific study objective.

- To me, it was not entirely clear how you estimate the prescription rates: is it mostly based on the structured input fields, or on the free text? And do you have any indication how much the number of mentions match prescriptions?

This concern is reinforced by your statement in the discussion that "It is important to note that change in frequency of clinical documentation of psychotropic medications is not necessarily correlated with changes in prescribing per se, and could reflect differences in the nature and frequency of EHR documentation between inpatient and community mental teams."

I feel this is an important limitation of your study, as this is the most important input to answer your research question on whether prescription has increased.

I would recommend to further specify in the Methods how mentions and prescriptions are related, or in other words how you derive your conclusions on this sub-question. And I would suggest to move/integrate the quoted part to the limitations.

- In my opinion, the observation that the decrease in clinical contacts during the early phase of COVID-19 was smaller than the decrease during Christmas and New Year period is not particularly relevant for the research question and now receives more focus than it deserves. It now is somewhat distracting in the presentation of your results and leads away from answering your research question. The same goes for similar observations regarding the prescription of medication.

- checklist point 10: In the results (and also in the Results in the abstract), you report about differences in clinical contacts and mode of contact between the three age groups, and also for gender. This is illustrated by Figure 4 and 5, which it seems now you base your conclusions on. Did you also statistically test whether these differences are significant? I see no such comparisons in Supplementary Table 1. If you did not, please report that the inferences are based on eyeballing, or include the necessary statistical tests.

That being said, when these comments are sufficiently addressed in the revision, I think the paper meets the standards to be accepted and will be a valuable contribution to the journal.

I hope my comments are clear and received in good order. I look

	forward to receive and review your revised manuscript.
REVIEWER	Christian Dalton-Locke University College London, UK
REVIEW RETURNED	29-Dec-2020

GENERAL COMMENTS	Thank you for the invitation to review this submission. The study is important and timely, and very well written. The Background and Discussion sections cover what I was expecting to find and includes interesting points and ideas for future research. My suggestions below are minor, the suggestions I would most like to see acted upon are number 4 and 9. Methods 1. Potentially expand on setting: What type of services are provided by SLaM (it is later described SLaM provides services for under 18s, 18-64 and 65+ but would be helpful to have this here instead, as well as type of service (e.g. includes substance misuse services?))? Is there anything particularly characteristic about the population demographics of Lambeth, Southwark, Croydon and Lewisham? 2. Defined variables: Expand on definition of 'in-person' contacts. I imagine this includes home visits and appointments at healthcare settings but would be good to add this, like list of included 'remote' contacts. 3. Defined variables: When deciding to group antipsychotics, was it considered to group clozapine on its own? I wonder if this may have been an interesting analysis given that it is usually prescribed for people with treatment resistant psychosis and the extra challenge of prescribing this medication during the social distancing measures given the necessary monitoring through in-person blood tests. (I see this was looked at in Supplementary Table 1) 4. Can a description of how a 'contact' was extracted from the records, and how this was extracted as a 'in-person' or 'remote' contact please be added. Definitions of in-person and remote contacts are provided but it is unclear how these definitions were used to extract these contacts from the data. Extraction of prescriptions was well described and referenced. Results: 5. Patients registered with mental health services: Is it possible to tell from the data whether there are changes in the number of 'new' people registered vs people 'continuously' registered? It would be interesting to see if there are changes in this between the index period and after March 2020 (I would hypothesise there would be a greater reduction in new people registered but greater reluctance to discharge people after 2020 vs before). 6. Contacts with mental health professionals: 7. Figure 4: Seems to be a typo on the y-axis, where '1k' should be '11k' 8. Figure 4: In addition to what the authors already comment on, there seems to be a second reduction in clinical events not long before July 2020 for both under 18s and over 65s but not for 18 to 65 year olds – am I correct in reading this figure this way? If so, could the authors please comment on this. 9. The efforts of the authors to provide an online interactive visualisation of the results is commendable. Can the link to this please be added somewhere in the main text rather than only in the Supplementary material.
--

VERSION 1 – AUTHOR RESPONSE

Reviewer: 1

Dr. Anjana Kavoor, Queensland Health

Comments to the Author:

The given study is on a topic of relevance given the current global situation. It aims to investigate the impact of the COVID-19 pandemic on the use of remote consultation and on the prescribing of psychiatric medications in various age cohorts. The paper is perspicuously written and easy to follow. The study has employed a sizable data set with a sound methodology and clear presentation of results. Results have confirmed the given research question of the impact on remote consultations, as one may presume given the limited alternatives during such a pandemic. It is interesting that the study ascertained that there was no significant difference in antipsychotic prescription during these consultations, although, like the authors rightly pointed out, this is only a part of the clinical interaction. It would also have been interesting to discern how many of those remote consultations were done over email or letters as opposed to video calls.

Given the limitations that the authors have already delineated, it is a well conducted study.

/*Thank you for your supportive comments. We analysed remote contacts which were attended by the patient and clinician simultaneously (i.e. phone and video calls). We agree that it would be useful to examine the frequency of written communication (e-mails/letters) but as these are not necessarily recorded contemporaneously (as e-mails and letters can be “time shifted”) we were not able to make reliable comparisons between frequency of audio/video and written communication on a weekly basis. We have updated the Methods section to clarify the approach used to extract data on in-person and remote contacts (Page 7) in response to point 4. of Dr. Christian Dalton-Locke’s comments (below).

However, we have extracted additional data comparing the frequency between phone and video consultations by each month to determine if there have been changes in the use of video call technology during the period of the study. We found that prior to the onset of the COVID-19 pandemic very few video consultations occurred, but there was a marked increase in video consultations after March 2020 which persisted subsequently (with video consultations accounting for around 30% of remote consultations).

We have provided additional charts on these data in Supplementary Figures 3 and 4. We have added text to the Results section as follows: “Before March 2020 very few video consultations were conducted. Following March 2020, the frequency of video consultations increased to around almost 30% of all remote consultations by July 2020 (Supplementary Figures 3 and 4).”*/

Reviewer: 2

Dr. Milou Feijt, University of Technology Eindhoven

Comments to the Author:

1. Is the research question or study objective clearly defined?

No

2. Is the abstract accurate, balanced and complete?

Yes

3. Is the study design appropriate to answer the research question?

No

4. Are the methods described sufficiently to allow the study to be repeated?

Yes

5. Are research ethics (e.g. participant consent, ethics approval) addressed appropriately?

Yes

6. Are the outcomes clearly defined?

Yes

7. If statistics are used are they appropriate and described fully?

Yes

8. Are the references up-to-date and appropriate?

Yes

9. Do the results address the research question or objective?

No

10. Are they presented clearly?

No

11. Are the discussion and conclusions justified by the results

Yes

12. Are the study limitations discussed adequately?

Yes

13. Is the supplementary reporting complete (e.g. trial registration; funding details; CONSORT, STROBE or PRISMA checklist)?

Yes

14. To the best of your knowledge is the paper free from concerns over publication ethics (e.g. plagiarism, redundant publication, undeclared conflicts of interest)?

Yes

15. Is the standard of written English acceptable for publication?

Yes

Dear authors,

I had the pleasure to review your article. I think it is an interesting paper, especially because of the access to the great amount of data and the novel way of presenting this visually.

I do have some comments, which I will elaborate upon below:

- As you can see from the checklist point 1 (and related 3 and 9), my main concern is about the definition of a clear and specific research question and study objective. To my opinion, it is well-defined in the abstract: "We investigated the impact of the pandemic on the use of remote consultation and on the prescribing of psychiatric medications.", as this is precisely what you have done. However, the study is introduced differently in the introduction, with a much less explicit research question/objective. "we analysed EHR data from a large provider of mental healthcare in South London (UK) to assess the impact of adopting remote consultation in response to the current COVID-19 pandemic. We tested the hypothesis that moving to remote consultation could be achieved without compromising the delivery of key components of secondary mental health care."

As a reader, this raises questions, as the objective is now formulated in much more general terms; it makes me wonder: what do you refer to with "the impact" and "key components of secondary mental health care". This is confusing, but more important, it also promises something that you can/do not deliver in the rest of the paper. I would advise to match the formulation in your introduction to the one in your abstract. This will also largely solve points 3 and 9 on the checklist, as your study design and results do match this more specific study objective.

/*Thank you for your supportive comments. We have updated the introduction section to align the formulation with that of the abstract (Page 5): "We analysed EHR data from a large provider of mental healthcare in South London (UK) to assess the impact of the current COVID-19 pandemic on rates of remote consultation and prescribing of psychiatric medications."*/

- To me, it was not entirely clear how you estimate the prescription rates: is it mostly based on the structured input fields, or on the free text? And do you have any indication how much the number of mentions match prescriptions?

This concern is reinforced by your statement in the discussion that "It is important to note that change in frequency of clinical documentation of psychotropic medications is not necessarily correlated with changes in prescribing per se, and could reflect differences in the nature and frequency of EHR documentation between inpatient and community mental teams."

I feel this is an important limitation of your study, as this is the most important input to answer your research question on whether prescription has increased.

I would recommend to further specify in the Methods how mentions and prescriptions are related, or in other words how you derive your conclusions on this sub-question.

/*We have added an additional figure to illustrate the breakdown of medication recording by structured input fields compared to data extracted from free text using NLP (Supplementary Figure 8). This illustrates that medication data are predominantly documented in unstructured free text in the SLAM EHR. Clinicians document medication prescribing in structured fields and free text as part of the intended treatment plan agreed with the patient, but this does not necessarily equal the number of prescriptions being dispensed by a pharmacy or being subsequently taken by the patient. It is likely that a proportion of prescriptions are not dispensed and a proportion of those which are dispensed are not taken, and that this may vary depending on the clinical setting or patient group receiving treatment. However, it is unlikely that psychiatric medications would be prescribed or dispensed in the absence of documentation as part of a treatment plan and so if there had been increases in the rates of medication prescribing, this would be correlated with an increase in the frequency of recording in the EHR. We have updated the limitations section of the Discussion to clarify this (Page 13)*/

And I would suggest to move/integrate the quoted part to the limitations.

/* We have now integrated the quote with the additional information above within the limitations section of the Discussion (Page 13).*/

- In my opinion, the observation that the decrease in clinical contacts during the early phase of COVID-19 was smaller than the decrease during Christmas and New Year period is not particularly relevant for the research question and now receives more focus than it deserves. It now is somewhat distracting in the presentation of your results and leads away from answering your research question. The same goes for similar observations regarding the prescription of medication.

/*The Christmas/New Year period at the end of December and beginning of January represents a period of planned mental healthcare service interruption. The reduction in clinical activity during this period serves as a useful comparator against which to examine the impact of unplanned potential or actual healthcare service interruption associated with the COVID-19 pandemic. However, we agree that our intention was not to specifically highlight the impact of the Christmas/New Year period on mental healthcare service delivery and this is not the aim of the present study. We have therefore amended the text to refer to this period by the calendar dates of "between December 2019 and January 2020".*/

- checklist point 10: In the results (and also in the Results in the abstract), you report about differences in clinical contacts and mode of contact between the three age groups, and also for gender. This is illustrated by Figure 4 and 5, which it seems now you base your conclusions on. Did you also statistically test whether these differences are significant? I see no such comparisons in Supplementary Table 1. If you did not, please report that the inferences are based on eyeballing, or include the necessary statistical tests.

/*We did not statistically test whether these differences are significant. To emphasise these inferences are based on visual data review we preface both paragraphs on Page 9 with these findings with "Visual inspection of Figure X indicates..."*/

That being said, when these comments are sufficiently addressed in the revision, I think the paper meets the standards to be accepted and will be a valuable contribution to the journal.

I hope my comments are clear and received in good order. I look forward to receive and review your revised manuscript.

Reviewer: 3

Dr. Christian Dalton-Locke, UCL

Comments to the Author:

Thank you for the invitation to review this submission. The study is important and timely, and very well written. The Background and Discussion sections cover what I was expecting to find and includes interesting points and ideas for future research. My suggestions below are minor, the suggestions I would most like to see acted upon are number 4 and 9.

/*Thank you for your supportive comments.*/

Methods

1. Potentially expand on setting: What type of services are provided by SLaM (it is later described SLaM provides services for under 18s, 18-64 and 65+ but would be helpful to have this here instead, as well as type of service (e.g. includes substance misuse services?))? Is there anything particularly characteristic about the population demographics of Lambeth, Southwark, Croydon and Lewisham?

/*We have expanded on the description of the mental healthcare services (Page 6) as follows: “[SLaM’s] service provision covers the following specialty groupings: Addictions; Behavioural and Developmental Psychiatry; Child and Adolescent Mental Health Services; Mental Health of Older Adults and Dementia; Mood, Anxiety and Personality; Psychological Medicine, and Psychosis. Its services are structured into three age groups: children and adolescents (under 18 years), working age adults (18 to 64 years) and older adults (65 years plus). SLaM’s catchment area varies considerably in terms of ethnic composition, education, urbanicity and area-level deprivation. Overall the SLaM catchment boroughs are representative of London as a whole in terms of age, gender, education and socioeconomic status.”*/

2. Defined variables: Expand on definition of ‘in-person’ contacts. I imagine this includes home visits and appointments at healthcare settings but would be good to add this, like list of included ‘remote’ contacts.

/*Please see below in response to point 4.*/

3. Defined variables: When deciding to group antipsychotics, was it considered to group clozapine on its own? I wonder if this may have been an interesting analysis given that it is usually prescribed for people with treatment resistant psychosis and the extra challenge of prescribing this medication during the social distancing measures given the necessary monitoring through in-person blood tests. (I see this was looked at in Supplementary Table 1)

/*We agree that clozapine recording rates over the past year may be particularly interesting given the need for regular blood monitoring in patients. For this reason, we performed a regression analysis of clozapine recording in our original analysis and identified a statistically significant increase in clozapine mentions in April 2020 alone when compared with the previous year (4th March 2019 to 1st March 2020) (page 10). Findings for all individual antipsychotic regression analyses are in Supplementary Table 1. We suggest reasons for the April increase in clozapine mentions in the Discussion (page 11), “The increase in clozapine mentions observed in April 2020 relative to the previous year may be due to the introduction of emergency guidelines regarding frequency of full blood count monitoring to manage possible disruptions during the initial period of travel and physical distancing restrictions. The increase in mentions of clozapine in April could reflect increased activity among clinicians creating treatment plans to navigate the challenges of managing full blood count monitoring for these patients.”*/

4. Can a description of how a ‘contact’ was extracted from the records, and how this was extracted as a ‘in-person’ or ‘remote’ contact please be added. Definitions of in-person and remote contacts are provided but it is unclear how these definitions were used to extract these contacts from the data. Extraction of prescriptions was well described and referenced.

/*We have added additional details to the methods section (Page 7) to explain how the data for contacts was extracted and the relationship to definitions as follows:

“Data on contacts with mental healthcare professionals were obtained from the *Events* input field in the EHR. Clinicians use the *Events* field to record the content and outcome of clinical appointments with patients. When an appointment for the patient to be assessed or reviewed by a clinician is scheduled, the clinician (or a healthcare service administrator) will create an *Event* and record the appointment date and time, whether the appointment was attended by the patient, and the *Event Type* (i.e. in-person or a remote contact). Data on contacts with mental healthcare professionals were defined as follows:

- (i) In-person contacts: appointments attended by a patient and clinician recorded as a “Face To Face” or “Group Contact” *Event Type*. The “Face To Face” *Event Type* refers to an appointment which is conducted with both the patient and clinician present in the same physical space which could be at a mental health service clinic/outpatient department or an alternative location such as the patient’s own home. The “Group Contact” *Event Type* refers to an appointment conducted in a group setting within the same physical space (i.e. multiple patients and one or more clinicians, e.g. as part of group psychological therapy).
- (ii) Remote contacts: defined as appointments attended by a patient and clinician recorded as a “Phone” or “Video (virtual) appointment” *Event Type*. We did not analyse data on written forms of remote contact between patients and clinicians (recorded as “Email”, “Letter”, “Mail” or “Short Message Text (SMS)” *Event Type*) as these methods of contact are not necessarily attended by the patient and clinician contemporaneously and the date of the recorded *Event* may not correspond with the date of the written communication being received by the patient.
- (iii) Did not attend (DNA) contacts: defined as any unplanned appointment cancellation (in-person or remote).”*/

Results:

5. Patients registered with mental health services: Is it possible to tell from the data whether there are changes in the number of ‘new’ people registered vs people ‘continuously’ registered? It would be interesting to see if there are changes in this between the index period and after March 2020 (I would hypothesise there would be a greater reduction in new people registered but greater reluctance to discharge people after 2020 vs before).

/*We have added a chart providing information on the number of patients newly accepted to the SLaM register each week to the Supplementary Material and the following text to Results section (Page 9): “The number of patients newly registered with SLaM services each week dropped between March and April followed by a modest recovery from May onwards (Supplementary Figure 2).”*/

6. Contacts with mental health professionals:

7. Figure 4: Seems to be a typo on the y-axis, where ‘1k’ should be ‘11k’

/*Thank you for pointing this out – we believe this was caused by a rendering issue in PDF generation of the article proof. We originally embedded the images as scalable vector graphic format (SVG) to preserve as much detail as possible. We have, instead, removed images from the main manuscript file and attached them as separate high-resolution bitmap TIFF files. This should ensure the figures accompanying the main article are reproduced correctly during the PDF rendering process but will remain high detail. For the supplementary material, we have attached the supplementary figures as a PDF file with bitmap images. We have also requested if the editorial team could provide supplementary files in *.docx format which would preserve the main article figures and supplementary figures in higher quality vector format than the PDF.*/

8. Figure 4: In addition to what the authors already comment on, there seems to be a second reduction in clinical events not long before July 2020 for both under 18s and over 65s but not for 18 to 65 year olds – am I correct in reading this figure this way? If so, could the authors please comment on this.

/*We agree there is a reduction for children under 18 compared to other age groups and this may represent the half-term period which occurs prior to the end of the school/academic year. This may be

associated with a brief reduction in clinical activity for school-age children. However, we do not feel we have sufficient data to make any inferences on these variations (which could simply be “chance” fluctuations) and so have not commented on them further in the manuscript.*/

9. The efforts of the authors to provide an online interactive visualisation of the results is commendable. Can the link to this please be added somewhere in the main text rather than only in the Supplementary material.

/*Thank you for your supportive comments regarding the interactive dashboard. We have included a link in the Methods section of the main manuscript in addition to the supplementary material (Page 8.*/

Reviewer: 1

Competing interests of Reviewer: None Declared

Reviewer: 2

Competing interests of Reviewer: None declared

Reviewer: 3

Competing interests of Reviewer: None declared

VERSION 2 – REVIEW

REVIEWER	Milou Feijt Eindhoven University of Technology
REVIEW RETURNED	18-Feb-2021

GENERAL COMMENTS	Thank you for your revised manuscript and responses to the reviewers. I am very pleased by the manner in which our comments are addressed, and I feel this has significantly benefitted the quality of the paper. In my opinion, the manuscript is now eligible to be accepted as is.
---

REVIEWER	Christian Dalton-Locke UCL, UK
REVIEW RETURNED	22-Feb-2021

GENERAL COMMENTS	Thank you for considering the reviewer comments and submitting a revised manuscript. I am satisfied by these responses and believe the study is of value to the journal's readership.
---